



# The impact of aerosol size-dependent hygroscopicity and mixing state on the cloud condensation nuclei potential over the Northeast Atlantic

Wei Xu[1,2], Kirsten N. Fossum[1], Jurgita Ovadnevaite[1], Chunshui Lin[1,2], Ru-Jin Huang[2], Colin O'Dowd[1], Darius Ceburnis[1]

[1]School of Physics, Ryan Institute's Centre for Climate and Air Pollution Studies, National University of Ireland Galway, Galway, Ireland.
[2]State Key Laboratory of Loess and Quaternary Geology and Key Laboratory of Aerosol Chemistry and Physics, Institute of Earth Environment, Chinese Academy of Sciences, Xi'an, China.

*Correspondence to*: Ru-Jin Huang(rujin.huang@ieecas.cn) and Colin O'Dowd (colin.odowd@nuigalway.ie)

**Abstract.** We present an aerosol cloud condensation nuclei (CCN) closure study over the Northeast Atlantic Ocean using six approximating methods. The CCN number concentrations ($N_{CCN}$) were measured at four discrete super-saturations (SS, 0.25, 0.5, 0.75 and 1.0%). Concurrently, aerosol number size distribution, sub-saturation hygroscopic growth factor and bulk $PM_1$ chemical composition were obtained at matching time resolution and after a careful data validation exercise. Method A used a constant bulk hygroscopicity parameter κ of 0.3; method B used bulk $PM_1$ chemical composition measured by an aerosol mass spectrometer (AMS); method C and D utilized a single size (165 nm) growth factor (GF) measured by humidified tandem differential mobility analyzer (HTDMA); method C utilized size-dependent GFs measured at 35, 50, 75, 110 and 165 nm; method E divided the aerosol population into three hygroscopicity modes (near-hydrophobic, more-hygroscopic and sea-salt modes) and the total CCN number in each mode was cumulatively added up; method F used the full size scale GF probability density function (GF-PDF) in the most complex approach. The studied periods included high biological activity and low biological activity seasons in clean marine and polluted continental air masses to represent and discuss the most contrasting aerosol populations.

Overall, a good agreement was found between estimated and measured $N_{CCN}$ with a linear regression slopes ranging from 0.64 and 1.6. The temporal variability was captured very well with Pearson's R value ranging from 0.76 to 0.98 depending on the method and air mass type. We further compared the results of using different methods to quantify the impact of size-dependent hygroscopicity and mixing state and found that ignoring size-dependent hygroscopicity induced overestimation of $N_{CCN}$ by up to 12 %, and ignoring a mixing state induced overestimation of $N_{CCN}$ by up to 15 %. The error induced by assuming an internal mixing in highly polluted cases was largely eliminated by dividing the full GF-PDf into three conventional hygroscopic modes while assuming an internal mixing in clean marine aerosol did not induced significant error.

## 1. Introduction

Aerosols play an essential role in climate change through direct and indirect effects (Twomey, 1977). The direct effect refers to the effect of aerosol absorbing and scattering incoming solar radiation, and the indirect effect refers to the activation of aerosol into cloud droplets thereby modifing cloud properties. Cloud condensation nuclei (CCN) is a subset of the aerosol population that can be activated into cloud droplets under certain water vapour supersaturation (SS).

Despite its essential role in regulating climate, the aerosol-cloud interaction have not been satisfactorily understood and resulting in the largest source of uncertainty in current climate models (IPCC, 2013). The accurate representation of natural aerosol is crucial for reducing the uncertainty in climate models (Carslaw et al., 2013). The CCN number concentration ($N_{CCN}$) is, therefore, crucial in characterizing aerosol-cloud interaction and their radiative impacts



The $N_{CCN}$ can be measured *in-situ* using a CCN counter (Roberts and Nenes, 2005), or estimated by the extrapolation of sub-saturation hygroscopicity based on Kölher theory (Köhler, 1936). Petters and Kreidenweis (2007) proposed a water activity

parametrization making it possible to describe aerosol hygroscopic growth and CCN activation with a single parameter, κ. Comparison between measured and extrapolated $N_{CCN}$ constitutes a so-called hygroscopicity - CCN closure. In the last few decades, intensive efforts have been made in achieving hygroscopicity - CCN closure (Cai et al., 2018; Cerully et al., 2011; Hansen et al., 2015; Hämeri et al., 2001; Hong et al., 2014; Jurányi et al., 2013; Kim et al., 2011). For example, Kawana et al. (2016) found that in urban area of Tokyo, the aerosol organic matter played a vital role in CCN activity, whereby the $N_{CCN}$ was

greatly underestimated if organic matter was assumed to be completely insoluble. Wex et al. (2010) compiled data from urban, rural and coastal areas for hygroscopicity - CCN closure, showing that the assumption of aerosol internal mixing lead to an overestimation of $N_{CCN}$ for continental aerosols, where less hygroscopic aerosol species dominated. A detailed exploration of hygroscopicity - CCN closure in a remote subarctic Sordalen mire site was conducted by Kammermann et al. (2010) and the results showed that ignoring particles mixing state did not impair CCN closure considerably but size-resolved hygroscopicity or

chemical composition information was essential for accurate estimation of $N_{CCN}$. Good et al. (2010) found good closure between sub-saturated hygroscopicity and CCN activity when nss-sulfate and ammonium were the major chemical species in terms of aerosol mass.

An alternative approach for $N_{CCN}$ estimation is to use chemical composition rather than subsaturated aerosol hygroscopicity because of the wider application of aerosol mass spectrometers (AMS) as compared to the HTDMA. Chemical composition can

be used to estimate water activity using various models, such as the Zdanovskii-Stokes-Robinson (ZSR) simple mixing rule (Stokes and Robinson, 1966; Zdanovskii, 1948) and thermodynamic models (Nenes et al., 1998; Topping et al., 2005; Zuend et al., 2011). Chemical composition- CCN closure has been successfully achieved in various environments, including marine (Martin et al., 2011), rural (Wu et al., 2013), urban (Ren et al., 2018), boreal (Hong et al., 2014) and free troposphere (Jurányi et al., 2010).

The above mentioned closure studies have provided great value in validating current small-scale thermodynamic and large-scale climate models. The discrepancies in certain closure studies revealed many factors that had not been taken into account, such as the change in solubility of organics (Petters and Kreidenweis, 2008), or liquid phase driven surface tension effects (Davies et al., 2019; Ovadnevaite et al., 2017; Petters and Kreidenweis, 2013; Ruehl et al., 2016). Successful closure studies suggested that it is possible to estimate the $N_{CCN}$ in large- scale models quite accurately. For example, Pringle et al. (2010) used

the global chemical composition distribution to model the global distribution of the hygroscopicity parameter κ.

Although chemical composition - CCN closure studies have been conducted more widely than the hygroscopicity - CCN closure studies, there are several benefits of using sub-saturation hygroscopicity to be further explored. Firstly, sub-saturation hygroscopicity measurements provide information about ultra-fine or Aitken mode particles which are not resolved in bulk chemical composition. Although size-resolved chemical composition data were presented in some locations by using a high

resolution aerosol mass spectrometer (AMS), it is difficult to achieve the same in remote environments where the mass concentration of submicron aerosols is usually too low to be size-resolved. Secondly, sub-saturation hygroscopicity measurements can reveal the aerosol mixing state. Chemistry-CCN closures often assume that aerosols are internally mixed, the assumption that can lead to over-estimation of CCN number (Ching et al., 2017, 2019). The HTDMA is one of the few instruments that can deliver the information of mixing state of submicron aerosol in near real-time. Thirdly, although the

hygroscopicity of inorganic salts is well established, the hygroscopicity of organics is still rather uncertain. The organic matter species are highly diverse and complex in the ambient environments, therefore, in many CCN closure studies a constant hygroscopicity of organics was assumed for simplification (e.g. Pringle et al., 2010). However, the hygroscopic properties of





organic matter were widely reported to be dependent on their molecular weight (Wang et al., 2019), oxidation level (Chang et al., 2010; Jimenez et al., 2009; Massoli et al., 2010; Nakao, 2017) and solubility (Petters and Kreidenweis, 2008). Freshly emitted
carbonaceous aerosols are often non-hygroscopic, while the aged secondary organic aerosols are more hygroscopic. The HTDMA can provide information on the hygroscopicity and mixing state of organic matter. Furthermore, the organic matter may transform into organic nitrate (Kiendler-Scharr et al., 2016) or organosulfate (Vogel et al., 2016), which have significant different hygroscopicity, thus introducing over- or under-estimation of the particle hygroscopicity. Lastly, it is well recognized that some of the organic compounds in the atmosphere can lower the surface tension of the droplets, thus leading to a decrease of
critical diameter at activation ($D_{crit}$). Such information is not necessarily provided by chemical composition measurement. Therefore, surface tension of water (0.072 N m$^{-1}$) is most often assumed. With the sub-saturation growth factor, the obtained $N_{CCN}$ was not affected by surface tension as long as $\kappa$ and surface tension were self-consistently applied in calculations (Petters and Kreidenweis, 2007), due to the surface tension used twice and cancelled out. There were few reported exceptions where the surface tension evolved with the relative humidity (RH) and liquid-liquid phase separation occurred leading to the surface
tension evolution during the growth of an organic enriched droplet (Liu et al., 2018; Ovadnevaite et al., 2017)

Unfortunately, a long-standing debate on the use of HTDMA for $N_{CCN}$ estimation is that the $\kappa$ derived from HTDMA measurements ($\kappa_{HTDMA}$) are generally lower than the $\kappa$ derived from CCN measurements ($\kappa_{CCN}$) for the same chemical species. For example, for $\kappa(NH_4)_2SO_4$ are defined as 0.53 and 0.61 from HTDMA and CCN measurement, respectively (Petters and Kreidenweis, 2007). Similarly, in both lab experiments and ambient measurements, the $\kappa_{CCN}$ were higher than $\kappa_{HTDMA}$ by about
~20 to 50% (Rose et al., 2010; Wu et al., 2013), while few previous studies found $\kappa_{HTDMA}$ and $\kappa_{CCN}$ agreeing well within 20% (Carrico et al., 2008; Duplissy et al., 2008). A meta-analysis compiled of HTDMA-CCN reconciliation over 10 sites revealed that in most of the sites the agreement was within the calculated error bars, however, there was a tendency of under-estimating $N_{CCN}$ using HTDMA, especially at low supersaturation (Whitehead et al., 2014).

Here we present CCN closure study from the Mace Head research station (MHD) using collocated aerosol number size
distribution measurements, $N_{CCN}$ measurements, sub-saturated hygroscopicity using HTDMA, aerosol bulk PM$_1$ chemical composition including refractory sea-salt by AMS and black carbon measurements by absorption photometer. Although several studies have investigated CCN and hygroscopic properties of aerosol in this region (Dall'Osto et al., 2010; Reade et al., 2006), the temporal coverage was limited, and no attempt has been made in estimating $N_{CCN}$ with hygroscopic properties of aerosols in various air masses and in different seasons. We compared the $N_{CCN}$ closure results by approximating both chemical composition
and hygroscopicity parameter and analysed the extent of deviations caused by different simplifications and assumptions that were commonly applied in modelling studies.

## 2. Methods

### 2.1 Site

Measurements were taken over the period from August 1$^{st}$ 2009 to November 12$^{th}$ 2009 and from 14$^{th}$ April 2010 to 23$^{rd}$ July
2010 at Mace Head atmospheric research station, located on the West coast of Ireland (www.macehead.org, last access: April 30$^{th}$ 2020) situated in the mid-to-high latitude of the North Atlantic Ocean. Meteorological parameters including solar radiation, wind speed, wind direction, relative humidity (RH) and pressure were also recorded over the measurement periods.

The aerosol population varied dramatically at MHD, ranging from polluted air masses advecting over the European Continent and the United Kingdom to the cleanest air masses advecting over North East Atlantic (Dall'Osto et al., 2010;





Ovadnevaite et al., 2014) and evolving by seasons, thus providing a unique opportunity to quantify the impact of simplified assumption in estimating $N_{CCN}$ in the above scenarios.

**2.2 Measurements**

**2.2.1 Aerosol number size distribution**

The aerosol number size distributions (20 to 500 nm) were measured using a scanning mobility particle sizer (SMPS). The SMPS
is comprised of a differential mobility analyser (DMA) and a condensation particle counter (CPC) which scans the full size range every 10 minutes. The SMPS system follows the particle physical properties measurement protocol developed by EUSAAR (http://www.eusaar.net/files/activities/na3.cfm, last access: Oct 12th 2020).

**2.2.2 CCN number concentration ($N_{CCN}$)**

The $N_{CCN}$ was measured using a Continuous-Flow Streamwise Thermal Gradient CCN counter (CCN-100, Droplet Measurement
Technologies, USA) described in Roberts and Nenes (2005). The CCN-100 operates at a flow rate of 0.5 L min$^{-1}$, which is separated into a wetted zero-air sheath flow and sample flow at a ratio of $10 \pm 0.3$ (sheath/sample). Aerosol is drawn into a temperature regulated wetted vertical column, which creates supersaturated conditions through its centerline proportional to the applied vertical temperature gradient along the column wall. The CCN-100 was operated at four discrete water vapor supersaturation of 0.25%, 0.5%, 0.75% and 1%, with a dwell time of 5 minutes per supersaturation (SS). Column SS was
calibrated using ammonium sulfate according to the methodology of Rose et al. (2008). Taking into account the length of this studies measurement period, and normal operational drift of the CCN-100 column SS, a SS uncertainty of ±0.03% was applied to the data (Schmale et al., 2017).

**2.2.3 Subsaturated hygroscopicity**

An HTDMA (Liu et al., 1978; Rader and McMurry, 1986; Swietlicki et al., 2008; Tang et al., 2019a) was used to measure the
GF-PDF. A detailed description of the HTDMA instrument operated at MHD can be found in previous studies (Bialek et al., 2012; Xu et al., 2020). Briefly, the HTDMA consists of two DMAs, a temperature triggered humidifier and a CPC. First, mono-disperse particles with certain electrical mobility were selected by applying a certain voltage to DMA-1. The particles were then humidified by a Gex-Tex membrane, and then scanned by the DMA-2 on which the varying voltage was applied corresponding to discrete particle electrical mobility diameter. By counting the particles at different voltages, GF-PDF was obtained over the
full size range from 35 to 165 nm. The GF was calculated by measuring the aerosol size distribution at a fixed RH. The GF is defined as the ratio of wet and dry diameters. To retrieve the GF-PDF from raw data and to correct the broadening of the DMA distribution, a piece-wise linear inversion algorithm was used (Gysel et al., 2009). In this study, the DMA-1 was held at an RH below 10 %, while the second DMA was set at an RH of 90 %. The electrical mobility diameters selected by the DMA-1 were 35, 50, 75, 110 and 165 nm. The main sample and sheath flow rates were 1 and 9 L min$^{-1}$, respectively. The operation and quality
assurance procedure followed the standard configuration and deployment recommended by the European Supersites for Atmospheric Aerosol Research (EUSAAR) network project (Duplissy et al., 2009).

It is important to note here that the definition of mixing state is arbitrary, although it was recently defined as the heterogeneous distribution of chemical species across the aerosol populations (Ching et al., 2017). Previous attempts of evaluating the impact of mixing state sometimes assumed chemical species being externally mixed with each other (e.g. Ren et al.
(2018)). While in the current study, the mixing state was defined as the distribution of hygroscopic species across the aerosol



population where the GF-PDF was practically divided by the bin of 0.01, and the binned GF-PDF was considered as the representation of the mixing state. Therefore, the method F always represented the real mixing state of the ambient aerosol.

### 2.2.4 Chemical composition

The $PM_1$ mass and chemical compositions, including organic matter (Org), ammonium ($NH_4$), non-sea-salt sulfate (nss-$SO_4$), nitrate ($NO_3$) and methanesulfonic acid (MSA) were measured using a high-resolution time-of-flight aerosol mass spectrometer (AMS, Aerodyne Research Inc., Billerica, MA) (DeCarlo et al., 2006). The $^{23}Na^{35}Cl^+$ ion signal at m/z 58 and a scaling factor of 51 were used to retrieve refractory sea-salt concentration (Ovadnevaite et al., 2012). The ion signal of $CH_3SO_2^+$ and $CH_3SO_3H^+$, which are related to MSA were used for quantification and calibration of MSA (Ovadnevaite et al., 2014). Regular calibrations were also conducted using ammonium nitrate an sulfate. The AMS was operating at vaporizer temperature of 650°C and 5 minute time resolution. The composition-dependent collection efficiency was applied (Middlebrook et al., 2012) to account for bouncing effects. Further details of the HR-ToF-AMS operations can be found in Ovadnevaite et al. (2014).

A multi-angle absorption photometer (MAAP, Thermo Fisher Scientific model 5012) was used to measure the mass concentration of black carbon (BC). The MAAP operated at a flow rate of 10 L min$^{-1}$. The MAAP measured the transmittance and reflectance of BC-containing particles at two angles to calculate the optical absorbance, as described by Petzold and Schönlinner (2004).

### 2.3 Data validation and quality assurance

Air masses that advected over MHD were categorized into clean marine and polluted continental types according to the hourly averaged BC concentration, wind direction and wind speed. The clean marine sector criteria were used following the previous studies where westerly air masses with BC concentration of less than 15 ng m$^{-3}$ convincingly separated anthropogenic impacted air masses (O'Dowd et al., 2014). The polluted continental sector was defined according to wind direction alone (Xu et al., 2020). The rest of the data were defined as "mixed" sector where either clean marine air masses were continentally modified or polluted continental air masses recirculated into marine sector wind direction ranges. The clean, polluted and mixed sectors were further divided by the level of oceanic biological activity further as "H" or "L" for high and low biological activity seasons, respectively. The duration of high biological activity periods classified according to prominent phytoplankton blooming periods (O'Dowd et al., 2004; Yoon et al., 2007) were longer in clean sector and shorter in polluted sector mainly due to synoptic scale conditions prevailing during different seasons. Table 1 summarizes the data selection criteria and their frequency of occurrence. The duration of Clean-H and Clean-L conditions were 237 hours and 68 hours, respectively, while the duration of Polluted-H and Polluted-L conditions were 98 and 345 hours, respectively.

The study used large overlapping data set of six different instruments and the data integrity was ensured by using a set of conservative processing criteria and taking advantage of a large initial dataset:

•

1. All of data were averaged to hourly resolution to match the resolution of various instruments due to different scanning time.

•

2. The data were manually checked to avoid any time stamp mismatch between individual data set.

•



3. It was mandated that the $N_{CCN}$ at higher SS was always higher than or equal to the $N_{CCN}$ at lower SS, otherwise, data were filtered out. In some cases, however, the $N_{ccn}$ only marginally increased at high SS, as the $D_{crit}$ moved in the tail of aerosol size distribution. For example, the $D_{crit}$ of sea-salt at 0.75% SS and 1% SS are about 28 nm and 23 nm, when assuming surface tension of water, and the gain in $N_{CCN}$ when summing up the aerosol size distribution from 28 nm to 23 nm would be minor, especially in low marine background events. Such small difference could mask itself within the measurement uncertainty.

•

4. The SMPS data were cross-checked by an independent condensation particle counter measurement (CPC, CPC3010, TSI). The ratio between $N_{10}$ (the total number of particle larger than 10 nm measured by SMPS) and $N_{cpc}$ (the total number measured by CPC) were calculated hourly. The upper envelope of the ratio was expected to be ranging between 1 to 1.1 to account for the frequent new particle formation at MHD. The SMPS data were corrected on a daily basis to make sure the 84.15th quantile of $N_{10}/N_{cpc}$ were ranging from 1 to 1.1. Considering the normal distribution of the $\log(N_{10}/N_{cpc})$, the value of 84.15th quantile of $\log(N_{10}/N_{cpc})$ represented the upper limit of 1 standard deviation over the mean value. The extra 10% of the uncertainty was allowed due to typical uncertainty of particle counting.

•

5. The total SMPS number was mandated to exceed that of $N_{CCN}$ to have meaningful comparison. For high particle number events ($N_{30} > 400$ cm$^{-3}$), the $N_{CCN}(0.75\%) < 1.1* N_{30}$ was considered as the upper credibility limit of activation, where $N_{30}$ represented the total particle number larger than 30 nm. For low number events ($N_{30} < 400$ cm$^{-3}$), the $N_{CCN} (0.75\%) < 1.2*N_{30}$ was considered as the upper limit. The different upper limits were taking into account a larger uncertainty in SMPS measurement at low total number concentration.

•

6. Only data periods lasting longer than 4 hours were used in further analysis to avoid transient events.

**Table 1: The total number of hours obtained in each air mass category after data validation.**

| Sector | Criteria | Level of biological acitvity | Month | Abbreviation | Duration (hours) |
|---|---|---|---|---|---|
| clean marine | BC < 15 ng $^{-3}$, WD from 190° to 300°, WS > 3 m/s | high (H) | May to Aug | Clean-H | 237 |
| clean marine | BC < 15 ng $^{-3}$, WD from 190° to 300°, WS > 3 m/s | low (L) | Oct to Apr | Clean-L | 68 |
| polluted continental | WD from 35° to 135 ° | H | May to Aug | Polluted-H | 98 |
| polluted continental | WD from 35° to 135 ° | L | Oct to Apr | Polluted-L | 345 |
| mixed | | H | May to Aug | Mix-H | 343 |
| mixed | | B | Oct to | Mix-L | 319 |



Apr

### 2.4 Uncertainty

The uncertainty of the aerosol number size distribution measured by the SMPS was normally less than 10% (Wiedensohler et al., 2017), the uncertainty of the HTDMA measurement was around 10%. The uncertainty of the AMS measurement was about 30-40%, which was mainly arising from the species dependent collection efficiency. However, only the relative mass fractions were used to calculate aerosol hygroscopicity arising from chemical composition. Therefore, we assumed the uncertainty of the AMS based κ to be lower than 20%. The RH of the air sample in the inlet of SMPS and AMS was always lower than 40%, while the inlet RH of the HTDMA was even lower, at ~10%. The uncertainty due to the inlet RH was minor because most of the inorganic aerosols undergo deliquescence at RH is greater than over 40%. For non-deliquescent organics, the increase in growth factor at 10% and 40% RH is typically limited to 10% of the electrical mobility sizes. The accuracy of GF-PDF is sensitive to the total number, therefore GF-PDF uncertainty was 20% for marine cases and even lower for polluted cases (Gysel et al., 2009). The accuracy in the CCN measurements was mainly determined by the accuracy of SS of the CCN counter column, which has an associated error on the order of $\pm$ 0.03% SS translating to approximately 10% uncertainty of measured $N_{CCN}$. Considering the uncertainties induced by interpolation between discrete sizes, any estimated $N_{CCN}$ within 30% of measured $N_{CCN}$ was believed to be reasonable closure. The relative root square error (RRSE) represented the relative error between the measurement and the estimated value, and it was calculated as

$$RRSE = \sqrt{\frac{\sum_i (p_i - y_i)^2}{\sum_i (y_i - \overline{y})^2}} \qquad (1)$$

where $p_i$ is the estimated value, $y_i$ is the measured value and $\overline{y}$ is the mean value of $y_i$.

### 3. Calculation of estimated $N_{CCN}$

### 3.1 Method A: Internally mixed $N_{CCN}$ calculation based on constant κ of 0.3

Method A applies a constant κ of 0.3 to all of the data regardless of aerosol composition. Method A is regarded as the simplest method that can be used for estimating $N_{CCN}$ and the κ of 0.3 has been suggested as the averaged κ over continental regions (Schmale et al., 2018). The $D_{crit}$ under certain SS was obtained by κ-Köhler theory (Petters and Kreidenweis, 2007).

$$D_{crit} = \sqrt[3]{\frac{4A^3}{27 \kappa ln^2 SS}} \qquad (2)$$

$$A = \frac{4 \sigma_{s/a} v_w}{RT \rho_w} \qquad (3)$$

where T is the temperature (298.15 K), R is the universal gas constant (8.315 J K$^{-1}$ mol$^{-1}$), $\rho_w$ is the density of water (997.1 kg m$^{-3}$), $M_w$ is the molar mass of water (0.018015 kg mol$^{-1}$), and $\sigma_{s/a}$ is surface tension in the interface of droplet and air (assumed to be 0.072 N m$^{-1}$).

Lastly, the number of CCN ($N_{CCN}$) was obtained by integrating the number size distribution from the $D_{crit}$ to its upper end of 500 nm.

$$N_{CCN}(SS) = \int_{D_{crit}}^{500nm} \frac{dN}{dlogD_p} dlogD \qquad (4)$$



**3.2 Method B: Internally mixed $N_{CCN}$ calculation based on the temporally resolved chemical composition**

The hygroscopicity parameter $\kappa_{chem}$ was calculated based on chemical composition using ZSR mixing rule, which assumed that the total water content of mixed particle is the sum of the water content of each species. Firstly, the $\kappa_{chem}$ was determined from the chemical composition as:

$$\kappa_{chem} = \sum_i \varepsilon_i \, \kappa_i \qquad (5)$$

where $\kappa_i$ and $\varepsilon_i$ are the hygroscopicity parameters and volume fractions of the individual chemical species in the mixture. The $\varepsilon_i$ was derived from the chemical composition obtained by AMS and MAAP:

$$\varepsilon_i = \frac{m_i/\rho_i}{\sum m_i/\rho_i} \qquad (6)$$

where the $m_i$ is the mass fraction, which converted to volume fractions based on the simplified ion-pairing scheme (Gysel et al., 2007; Wu et al., 2013), and $\rho_i$ is the density of each chemical. The $\kappa_i$ and $\rho_i$ used in this study are summarised in Table 2.

The $N_{CCN}$ was then calculated using equation (4) and (2) by replacing $\kappa$ with $\kappa_{chem}$.

**Table 2: The $\kappa$ values and densities of the chemical species considered in the study.**

| species | $\rho$(density, kg m$^{-3}$) | $\kappa$ |
|---|---|---|
| (NH$_4$)$_2$SO$_4$ | 1769 | 0.58 |
| NH$_4$HSO$_4$ | 1780 | 0.56 |
| H$_2$SO$_4$ | 1830 | 0.68 |
| NH$_4$NO$_3$ | 1720 | 0.48 |
| sea-salt | 2165 | 1.12[a] |
| Organics | 1400 | 0.1 |
| MSA | 1481 | 0.6[b] |
| BC | 1650 | 0 |

**Note:** [a] the value was adapted from Zieger et al. (2017). [b] the value was adapted from Fossum et al. (2018) and Tang et al. (2019b).

**3.3 Method C: Internally mixed $N_{CCN}$ calculation based on temporally resolved single growth factor**

Method C is similar to method B, while the $\kappa_{chem}$ was replaced by $\kappa_{HTDMA}$, which was obtained as follows:

$$\kappa_{HTDMA} = (GF_{mean}(Dp)^3 - 1)\left(\frac{exp\left(\frac{A}{D_p \cdot GF_{mean}(Dp)}\right)}{RH} - 1\right) \qquad (7)$$

where the RH is the relative humidity of DMA-2 of HTDMA. and the $GF_{mean}$ (Dp) is the mean GF of the particle diameter (Dp), which was obtained by integrating the GF-PDF c(GF, 165 nm) (Gysel et al., 2009) from 0.8 to 2.5 with a $\Delta$GF of 0.1:

$$GF_{mean}(Dp) = \int_{0.8}^{2.5} G F \cdot c(GF, Dp) dGF \qquad (8)$$

In method C, only the Dp = 165 nm was used as representing the accumulation mode, and other sizes were assumed to have the same $\kappa_{HTDMA}$.





### 3.4 Method D: Internally mixed $N_{CCN}$ calculation based on size-dependent growth factor

Method D used the size-dependent growth factor instead of a single growth factor and the measured Dps were 35, 50, 75, 110
and 165 nm. Linear interpolation was used between these sizes to obtain a full-size range measured by SMPS. To note that the
particle diameters larger than 165 nm were assumed to have a growth factor equal to that of 165 nm diameter particles, and the
sizes smaller than 35 nm were assumed to have GF that of 35 nm.

### 3.5 Method E: $N_{CCN}$ calculation based on three hygroscopic growth modes

In method F, the GF-PDF were divided into near-hydrophobic mode (1 < GF< 1.3), more-hygroscopic mode (1.3 < GF <1.85)
and sea-salt (1.85 < GF). The number fraction (nf) and GF of these modes were calculated as follow:

$$nf_{near-hydrophobic} = \int_{1}^{1.3} c(GF,D)dGF \qquad (9)$$

$$GF_{near-hydrophobic} = \frac{1}{nf_{near-hydrophobic}} \int_{1}^{1.3} GF \cdot c(GF,D)dGF \qquad (10)$$

$$nf_{more-hygroscopic} = \int_{1.3}^{1.85} c(GF,D)dGF \qquad (11)$$

$$GF_{more-hygroscopic} = \frac{1}{nf_{more-hygroscopic}} \int_{1.3}^{1.85} GF \cdot c(GF,D)dGF \qquad (12)$$

$$nf_{sea-salt} = \int_{1.85}^{2.5} c(GF,D)dGF \qquad (13)$$

$$GF_{sea-salt} = \frac{1}{nf_{sea-salt}} \int_{1.85}^{2.5} GF \cdot c(GF,D)dGF \qquad (14)$$

And then the $D_{crit}$ of each mode was calculated by Equation (4). Finally, the $N_{CCN}$ was the sum of the activated number of
particles in three modes:

$$N_{CCN} = \sum_{i} (\int_{D_i}^{500nm} nf_i(Dp) \frac{dN}{dlogD_p} dlogD) \qquad (15)$$

where $nf_i(Dp)$ is the number fraction of each mode at Dp size.

### 3.6 Method F: $N_{CCN}$ calculation based on temporally resolved growth-factor probability density function (GF-PDF)

The calculation of $N_{CCN}$ based on sub-saturation hygroscopicity followed the method described in Kammermann et al. (2010).
This approach was considered as the most detailed CCN estimation which takes the mixing state and size-dependent
hygroscopicity into account. The approach calculates the number fraction of CCN ($f_{CCN}$) based on the GF-PDF, which was
obtained as follows:

$$f_{CCN}(SS, D_p) = \int_{GF_{crit(SS,Dp)}}^{GF=2.5} c(GF, D_p)dGF \qquad (16)$$

where $GF_{crit}(SS, D_p)$ is the smallest GF that required when a size of $D_p$ particle activated under SS. The $GF_{crit}(SS, D_p)$ is
obtained by using Equations (7), (2) and (3). The full GF-PDF over SMPS size ranges was obtained by interpolating the GF-PDF
over the five measured electrical mobility diameters using linear interpolation.

The estimation of $N_{CCN}$ is then obtained by integrating the number size distribution weighted with the activated number
fraction $f_{CCN}(SS, D_p)$:


$$N_{CCN}(SS) = \int_{20nm}^{500nm} f_{CCN}(SS, D_p) \frac{dN}{dlogD_p}(D_p) dlogD_p \qquad (17)$$

A schematic graph summarizing key principles of each method is given in Fig. 1, where the first row of panels exhibits the types of hygroscopicity parameter κ used and the second row of panel visualizes how the $N_{CCN}$ were calculated using aerosol number size distribution. The defining features and simplifications of each method regarding to size-dependent hygroscopicity and mixing state are given in Table 3.

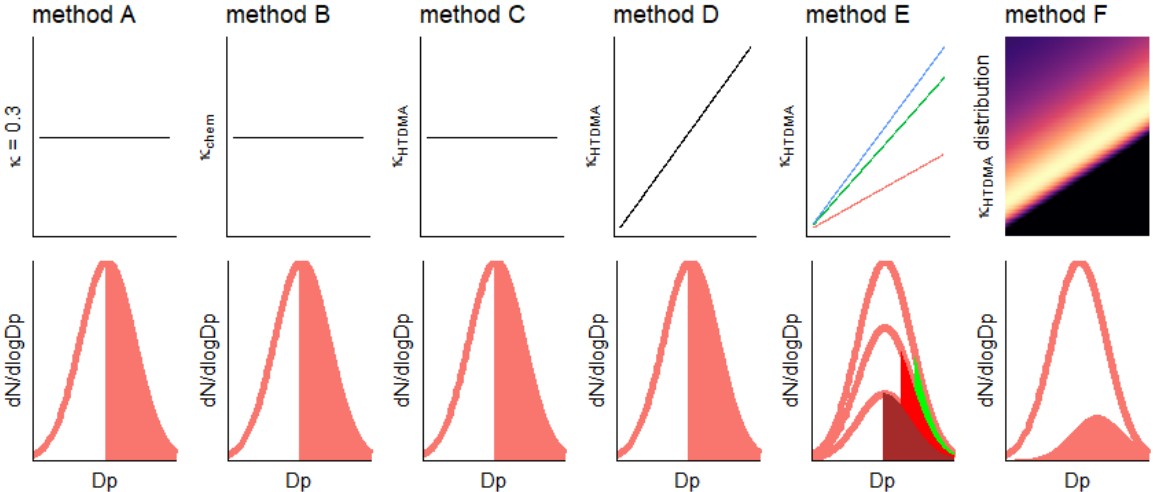

**Figure 1: Schematic illustration of different methods used for $N_{CCN}$ estimation. Method A: constant κ of 0.3; method B: size-independent κ derived by bulk PM₁ chemical composition; method C: size-independent κ derived by hygroscopicity growth factor(GF) at 165 nm; method D: size-dependent κ derived by hygroscopicity growth factor; method E: size-independent κ and number fractions of near-hydrophobic, more-hygroscopic and sea-salt modes; method F: size-dependent activation ratio based on GF-probability density function (GF-PDF)**

**Table 3: The simplification and assumption of $N_{CCN}$ estimation methods.**

| Method | Hygroscopicity proxy | Mixing state assumption | Size dependent hygroscopicity | Temporal variation |
|---|---|---|---|---|
| A | constant κ of 0.3 | internal mixing | no | no |
| B | bulk PM1 chemical composition | internal mixing | no | yes |
| C | mean GF | internal mixing | no | yes |
| D | mean GF | internal mixing | yes | yes |
| E | number fractions of hygroscopicity modes | quasi external mixing | yes | yes |
| F | GF-PDF | external mixing | yes | yes |



## 4. Results and discussion

### 4.1 Measured CCN concentration and aerosol size distribution

The aerosol number concentrations and $N_{CCN}$ varied with air mass types and seasons. Fig. 2 shows the aerosol number size distributions for each type of air mass. The Clean-H and Clean-L showed typical bimodal distribution of marine aerosol and similar to the open ocean nucleation event category (Dall'Osto et al., 2010; O'Dowd et al., 2010). The Aitken mode peaked at a dry electrical mobility diameter of 30 to 50 nm and accumulation mode peaked at 100 to 200 nm. However, the size distribution in Polluted-H and Polluted-L showed broad distributions centred at 50-60 nm, but in Polluted-L, the relative contribution of accumulation mode was more significant due to the increased cloud processing during winter. Fig. 3(a) summarizes the $N_{CCN}$ for the given SSs for each type of air mass, to note that the SS of 1% data was not included, as it was partially unavailable. Overall, the measured $N_{CCN}$ in Polluted-L was the highest (with a median concentration of 1164 cm$^{-3}$ at SS of 0.25% and 1800 cm$^{-3}$ at SS of 1%) which was attributed to the high total particle number concentration (the median particle number with a diameter larger than 30 nm was 3156 cm$^{-3}$) consistent with continental sources and enhanced emission. Polluted-H also showed remarkably high $N_{CCN}$ and aerosol number concentrations, with the $N_{CCN}$ median of 1182 cm$^{-3}$ at SS of 1% and median number particle of $N_{30}$ of 1499 cm$^{-3}$. In comparison to the polluted sector, low number concentrations were found in clean sector, reflecting the clean marine conditions over the Northeast Atlantic Ocean selected by the sector criteria (Methods section). For the clean sector, seasonal variation was also observed. Specifically, the Clean-H showed higher $N_{CCN}$ than Clean-L (for SS of 0.5%, median $N_{CCN}$ values are 171 and 114 for Clean-H and Clean-L, respectively), although the bulk hygroscopicity of Clean-L aerosol was higher because of the higher contribution of sea-salt. The mean $\kappa_{chem}$ were 0.63 and 0.79, and the $\kappa_{HTDMA}$ values for 165 nm particles were 0.45 and 0.56 for Clean-H and Clean-L, respectively, see Table S2 and Table S3. The $N_{CCN}$ difference between Clean-H and Clean-L could be explained by the fact that the aerosol number concentration was high during Clean-H (the median numbers of particles over 30 nm are 322 cm$^{-3}$ and 206 cm$^{-3}$ for Clean-H and Clean-L, respectively), due to the prominent biological activity over the ocean, leading to higher concentration of condensible vapour and secondary aerosol formation. The mixed sector showed intermediate values of total particle concentrations and size distribution that fell between the clean sector and polluted sector patterns. Mixed-H showed greater similarity in number size distribution to Clean-H, while the Mixed-L was closer to Polluted-L in number size distribution due to the prevailing winds and synoptic scale circulation in the low biological activity periods being predominantly influenced by continental air advection.

The activation ratios (AR) at varying SS for each category are shown in Fig. 3(b). The AR in clean sector were generally higher than that of the polluted sector. Clean-L was characterised by higher AR than Clean-H at SS of 0.25%, while at SS of 1%, the AR was higher in Clean-H, which is consistent with chemical composition and size-dependent hygroscopicity (Table S2 and Table S3). In Clean-H, the smaller particles exhibited higher hygroscopicity, while in Clean-L, it was the larger particles that were more hygroscopic. This pattern is attributed to the presence of Aitken mode near-hydrophobic particle in Clean-L, as highlighted in the previous study (Xu et al., 2020). The AR of less than 100% in Clean-L suggested that treating winter time marine aerosol as pure sea-salt or their mixture with nss-SO$_4$ will likely induce an over-estimation of $N_{CCN}$ at higher SS. Further research is needed to determine the source of such near-hydrophobic Aitken mode particles. The reduced AR in Polluted-L sector was consistent with the past studies showing that anthropogenic-related organics significantly reduced the particle hygroscopicity (Duplissy et al., 2011; Wu et al., 2013), while the higher AR in Polluted-H lend support the idea that enhanced particle aging by photochemical reactions druing spring, summer and autumn, making them more hygroscopic and CCN active at low SS.





The bulk PM$_1$ chemical composition and derived $\kappa_{chem}$ for each category is shown in Table S2. The concentrations of BC in clean marine sector were 7.1 and 6.4 ng m$^{-3}$ for HB and LB seasons, respectivel, reaffirming the representativeness of clean marine air masses.

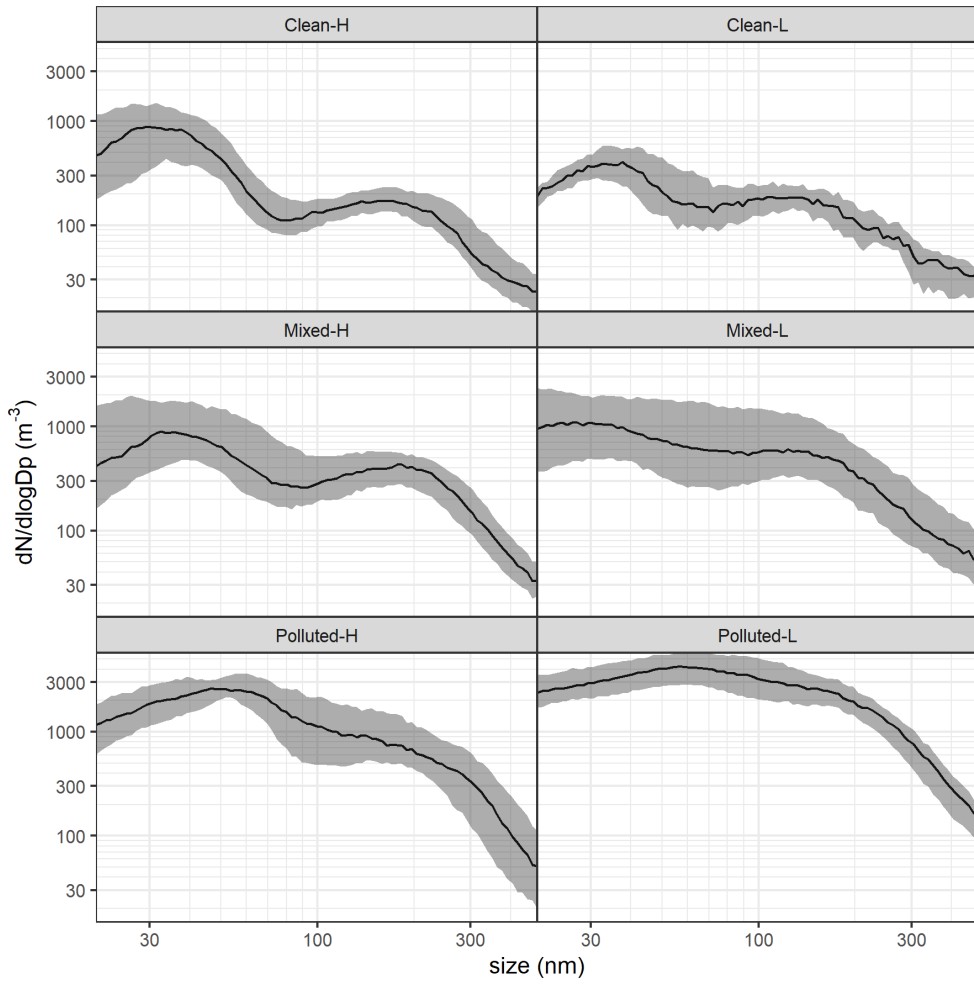

345

**Figure 2: The aerosol number size distribution for different types of air masses. The solid lines represent the median number concentration (dN/dlogDp), and the shaded area represents the 25$^{th}$ to 75$^{th}$ percentile.**





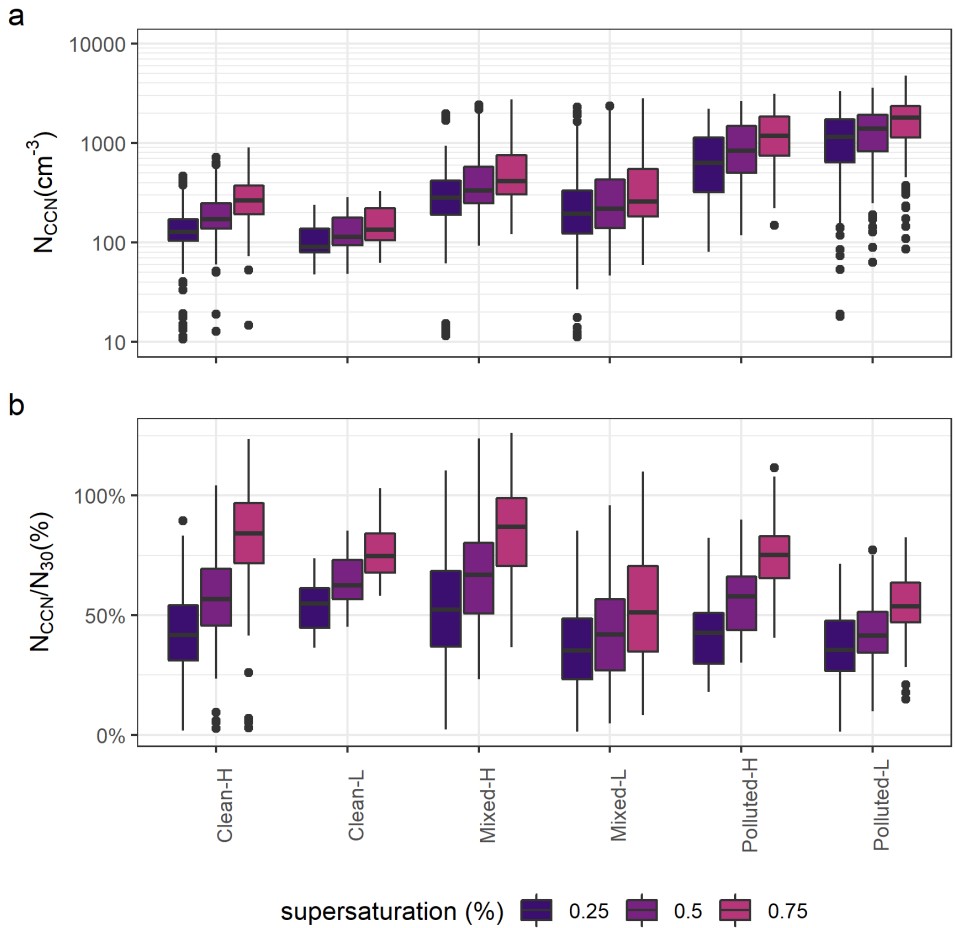

**Figure 3:** (a) The $N_{CCN}$ and (b) the activation ratio($N_{CCN}/N_{30}$) at given supersaturations (SSs) for different type of air mass, the lines represent the median, boxes represent 25th to 75th quantile and whiskers represent 1.5 interquartile and the points represent outliers . The colour codes represent the level of supersaturations. Note the logarithmic Y axes in (a).

**4.2 CCN closure**

The measured $N_{CCN}$ and estimated $N_{CCN}$ were compared by different air mass back trajectory sectors and seasons. The closure was considered as achieved when measured $N_{CCN}$ and estimated $N_{CCN}$ agreed within the range of measurement uncertainty (refer to Section 2.4). The sector specific closure results are shown in Fig. 4&-5. Overall, the estimated and measured $N_{CCN}$ agreed well and were highly correlated with the Pearson's R ranging between 0.85 and 0.99. The slope of regression line (with a fixed intercept of zero) larger than 1.0 suggested over-estimation, while the slope smaller than 1.0 suggested under-estimation.

Here we show and discuss the results for Clean and Polluted air mass back trajectory sectors, while the closure results of Mixed-H and Mixed-L are shown in Fig. S1. Overall, the closure results using different methods are similar within each sector. This points to the fact that for $N_{CCN}$ estimation, the effect of hygroscopicity or chemical composition are weaker than the particle size (Dusek et al., 2006; Wang et al., 2018). The statistical parameters estimated $N_{CCN}$ are given in Table S1. The normalised frequency of distribution of critical diameter by method A and method D is given in Fig. S2 and Fig. S3.



### 4.2.1 Clean sector

Hygroscopicity-CCN closure or chemistry-CCN closure in marine environments are few and apart due to the inaccessibility to

365     the marine environment. By contrast, well geographically located sites like MHD, offer ample opportunities for the systematic study of the marine atmosphere by carefully selecting observation periods. In this study, the clean sector represented the air masses advected across the North Atlantic ocean without anthropogenic impact, and setting up the regional aerosol background entering the European Continent. The low number concentration (See Section 4.1) and low black carbon mass concentration suggested the least anthropogenic impacts. The accumulation mode Clean-L aerosol were mainly composed of $nss-SO_4$ and sea-

370     salt, as suggested by the chemical composition (Table S2) and hygroscopicity measured at 110 and 165 nm (Fig. S5), while the Aitken mode aerosol consisted of a fraction of near-hydrophobic particles (Fig. S5), which was not discerned by the bulk chemical composition due to negligible mass (Xu et al., 2020). As shown in Fig. 4, the Person's R values for the clean sector ranged from 0.7 to 0.97, depending on the type of air mass and SS. In the Clean-L, sea-salt was the major contributor to the $PM_1$ mass loadings, leading to a high averaged of $\kappa_{chem}$ of 0.78. On the contrary, $nss-SO_4$ and organics accounted for a larger fraction

375     in Clean-H. The averaged BC concentrations of 6.5 ng m$^{-3}$ confirmed the cleanness and representativeness of the filtered clean sector data.





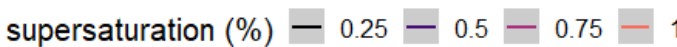





**Figure 4: Comparison of measured $N_{CCN}$ against the estimated $N_{CCN}$ under different SS using each method in Clean-L and Clean-H periods. The colour codes represent SS, equation of regression line, Pearson correlation efficient and relative root square error (RRSE) are shown in each panel. The dashed lines represent ±30% uncertainty ranges. To note that axes are in logarithmic scales.**

For the Clean-L, the slopes of regression lines of method A ranged from 0.76 to 1.4, and the Pearson's R values were 0.70 to 0.97, the slopes of method A were among the lowest, which was expected as either $\kappa_{HTDMA}$ or $\kappa_{chem}$ values were higher than 0.3, due to the high contribution of sea-salt to the aerosol mass (Table S2 and Table S3). Compared to method F, method A showed about 10 to 15% under-estimation at SS from 0.25 to 0.75%. Method B and method C also showed significant over-estimation at SS over 0.5% and the correlations were the highest when SS were 0.5 or 0.75% due to the fact that the $D_{crit}$ was moving towards the tail of SMPS spectrum with increasing SS, where the change of $D_{crit}$ did not produce a considerable change in $N_{CCN}$. Similar results between method B and C suggested minor difference between $\kappa_{chem}$ and $\kappa_{HTDMA}$ in Clean-L. The difference in $N_{CCN}$ between method C and method D was arising from size-dependent hygroscopicity. The $N_{CCN}$ obtained by method D was generally lower than that of method C by 3% to 19%, as shown in Table S3 with the larger particles exhibiting higher hygroscopicity. The reduced hygroscopicity in smaller sizes were consistent with our previous observation of near-hydrophobic Aitken mode particles in winter time clean marine air masses (Xu et al., 2020).

No statistically significant difference was found between method D and method E with regard to slopes and the RRSE, suggesting that the simplification of full GF-PDF into near-hydrophobic, more-hygroscopic and sea-salt modes was a good representation of aerosol mixing state. Similar results were obtained between method C, D and E at any given SS demonstrating minor impact of mixing state to $N_{CCN}$ closure. This is probably due to the fact that the although aerosol particles were externally mixed sea-salt and nss-SO$_4$, both of them were pretty hygroscopic. Consequently, if the species were not hygroscopic the impact of neglecting the mixing state would be significant.

For the Clean-H sector, the Pearson's R values ranged from 0.76 to 0.98, suggesting that most of the variability of measured $N_{CCN}$ was captured well. The slopes ranged from 0.58 to 1.1 and increased with increasing SS. The Pearson's R values were high as high as 0.96 using method A and the slopes ranged from 0.58 to 0.8 suggesting significant under-estimation by using a constant $\kappa$ of 0.3. The $N_{CCN}$ using method B was substantially higher than method A by 10% to 40%, because of high $\kappa_{chem}$ values (averaged $\kappa_{chem}$ of 0.63). No statistical difference between method C, D, E and F was found in terms of slopes, R values and the RRSE, the similarity being consistent with the internal mixing of Clean-H aerosol. As shown in Fig. S5, the size-resolved GF-PDF of the Clean-H aerosols were mainly composed of more-hygroscopic mode which can be the nss-sulfate or the mixture of sea-salt and organics. The steep activation curve derived by GF-PDF also confirmed the minor difference between method B and C (Fig. S4).

The results in clean sector were different with other closure studies in marine environments. For example, a closure study conducted during RHaMBLe Discovery Cruise by Good et al. (2010) found the discrepancy between $\kappa_{HTDMA}$ and $\kappa_{CCN}$. They found consistent under-estimation of CCN activity by using aerosol composition and sub-saturation hygroscopicity. A CCN closure study in summer high Arctic during Arctic Summer Cloud Ocean Study (ASCOS) found the calculated CCN was always higher than the measured CCN at the SS of 0.73% and 0.41% (Martin et al., 2011). Ovadnevaite et al. (2011) found that marine primary organic aerosol exhibited low GF and high CCN activation ratio simultaneously and hypothesized that the formation of marine hydrogel was responsible for the measured dichotomy. Similarly, Ovadnevaite et al. (2017) found underestimation of $N_{CCN}$ by using chemical composition by the an order of magnitude, and liquid-liquid phase separation was considered as the main driver of surface tension lowering. On the contrary, Mochida et al. (2011) calculated CCN efficiency spectra from hygroscopic growth and it agreed reasonably well with measured CCN spectra without lowering of the surface tension. When comparing to those studies, there were few important differences in the current study: 1) the sea-salt component was retrieved





with our method, which was not measured by some of the above studies; 2) our conservative data selection approach aimed at removing anthropogenic influence by using the threshold of BC of 15 ng m$^{-3}$, while in some of the above mentioned studies,

some impact of anthropogenic perturbation might have influenced results; 3) lastly, an apparent dichotomy during significant marine organic matter events would violate closure. The organic matter events are regularly observed at MHD with oraganic matters (OM) concnetration approaching 3-4 μg m$^{-3}$ and lasting up to 24 hours (Ovadnevaite et al., 2011, 2017). The above events can be dominated by eith primary and/or secondary marine biogenic sources and most importantly characterized by low hygroscopicity particles, which however readily activated to CCN and forming cloud droplets (Ovadnevaite et al., 2011). The

violation of closure depends on the specific κ value of OM during the events, and the mixing ratio with sea-salt (Vaishya et al., 2013) and whether or not OM is lowering the particle surface tension (Ovadnevaite et al., 2017). In this paper we used surface tension of water throughout the study, which does not apply during the OM events, and as such the detailed expolration of OM events is beyond the scope of the current study. Furthermore, the OM events were extensively covered in the above studies.

Our current methodological approach did not allow the calculation of $\kappa_{CCN}$, which required size-resolved CCN measurement,

making it challenging to compare $\kappa_{CCN}$ with $\kappa_{HTDMA}$ or $\kappa_{chem}$ (and concurrently advancing understanding of OM events). However, by comparing measured and estimated $N_{CCN}$, we conclude that using constant κ of 0.3 is likely to induce significant under-estimation of $N_{CCN}$, and using the HTDMA and AMS enabled to predict $N_{CCN}$ accurately well most of the time, at least at MHD.

### 4.2.2 Polluted sector

Polluted sector represents the aerosol population in the polluted air masses advected across Ireland from the UK or European

continent with higher mass loadings of chemical species and higher number concentrations, leading to a higher $N_{CCN}$. Aerosols in polluted sector were rather externally mixed, which consisted of near-hydrophobic mode and more-hygroscopic mode (Fig. S5), and led to the flatter activation curves (note panel of Fig. S4 for clarity). The near-hydrophobic mode was mainly composed of anthropogenic organic matter or BC and their mixture with nss-SO$_4$ and NH$_4$, and the more-hygroscopic mode mainly composed of secondary inorganic species. The BC concentrations (290 and 410 ng m$^{-3}$ for Polluted-H and Polluted-L) were nearly an order

of magnitude higher than those in the Clean sector. Polluted-L also showed a high mass contribution of organics. Higher BC and organic matter in Polluted-L suggested significant contribution of biomass and fossil fuel combustion in winter. Fig. 4 shows the $N_{CCN}$ closure of Polluted-H and Polluted-L sectors.





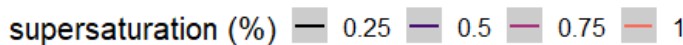





**Figure 5: Comparison of measured N$_{CCN}$ against estimated N$_{CCN}$ under different SS using each method in Polluted-L and Polluted-H. The colour codes represent SS, equation of regression line, Pearson correlation efficient and relative root square error (RRSE) are shown in each panel. The dashed lines represent 30% uncertainty ranges. To note that axes are in logarithmic scales.**

For the Polluted-L, the Pearson's R values ranged from 0.74 to 0.97, suggesting that the temporal variability of N$_{CCN}$ were well explained by either chemical composition or sub-saturation hygroscopicity. The slopes were the highest at 0.75% SS, while the Pearson's R values were highest at 1% SS, RRSE values were lowest at 0.75% SS. The highest Pearson's R values obtained

at 1% SS was likely the combined result of hygroscopicity and aerosol number size distribution. The obtained D$_{crit}$s were between 40 and 50 nm (Fig. S3), which was within the peak of aerosol number size distribution, where the uncertainty of aerosol number size distribution was the lowest. As shown in the method A panel of Fig. 4, using a constant κ of 0.3 results the steepest slope, for example, the slope of 1.5 at 0.5% and 1% SS, which is higher by over 30% than obtained by method F. That indicated the potential over-estimation of 30% to 50% by using κ of 0.3 in highly polluted air masses. Method A and method B showed

very similar results, due to similar κ values used in the calculation. As shown in Table S2, the κ$_{chem}$ was 0.31 ± 0.097 during the Polluted-L and the mean κ$_{chem}$ (0.30 ± 0.1) was larger than κ$_{HTDMA}$ for 165 nm particles (0.17 ± 0.07), suggesting the possible over-estimation by using the assumed κ$_{org}$ of 0.1. In a previous sub-saturated hygroscopicity and chemical composition study (Xu et al., 2020), the closure was achieved during wintertime between GF$_{HTDMA}$ and GF$_{AMS}$, suggesting that using κ instead of GF will substantially increase an apparent error, due to their cubic relationship, as shown in Equation (7). Moreover, the selected κ

values for N$_{CCN}$ closure were usually in the upper range of hygroscopicity of specific chemical species, leading to the larger discrepancies between κ$_{chem}$ and κ$_{HTDMA}$ (Table S2 and Table S3).

In the Polluted-H sector, the slopes ranged from 0.6 to 1.1, and the Person's R values ranged from 0.93 to 0.97. The slopes and correlations were the lowest at low SS. The method A showed the highest slopes of regression, with the majority of the slopes exceeding 1 except at 0.25% SS. The differences between method A and B were minor, suggesting the influence of

replacing κ$_{chem}$ by κ$_{HTDMA(165nm)}$ was minor, and consistent with our previous study on the chemical composition-hygroscopicity closure (Xu et al., 2020). The slopes of the method C (0.71 to 1.1) were lower than that of the method B (0.79 to 1.1), indicating the assumption made about the hygroscopicity being homogeneous across the sizes contributed to over-estimation of N$_{CCN}$ by less than ~10%. Method D had the slopes closest to 1, suggesting the best achieved closure and the minor contribution of Δκ or surface tension reduction. The N$_{CCN}$ results obtained by method E were lower than method D, but higher than method F.

CCN closure studies in polluted area were attempted in the past and the results varied rather significantly: Ervens et al. (2010) used simplified assumptions of organics aerosol from six locations and the slopes of regression ranged from 0.2 to 7.9, and the locations further away from fresh emission source are more reliable, similar to the closure studies at the high alpine site and MHD (Jurányi et al., 2010). Compare to the aforementioned studies, the site in the current study, MHD, is also far away from fresh anthropogenic emission sources (O'Connor et al., 2008), making MHD less influenced by the mixing state. A more

recent study by Schmale et al. (2018) compiled CCN closure by using chemical composition, particle number size distribution and CCN measurement from 12 sites on 3 continents. When applying the simple κ-Köhler theory and assuming internal mixing and ignoring the size -dependency, the slopes ranged from 0.87 to 1.37, and κ$_{org}$ worked reasonably well in their study. However, there were a few differences in our method versus the Schmale et al. (2018) study, in which MHD data were also included: 1) the temporal coverage was different; 2) Schmale et al. (2018) selection of κ of chemical species was different from this study. The

slope of regression lines were ~1.14 and Pearson' R value were ~0.97, suggesting that the selection of higher κ values might induce a slight over-estimation but still result in good correlation. However, due the variability of aerosol in MHD and different temporal coverage of the two studies, the difference in slopes not only resulted from the selection of κ, but also the seasonal





coverage of the data. Overall, the reasonable closure at MHD confirmed that data quality and aerosol number size distribution are essential to accurate $N_{CCN}$ estimations.

To conclude, using κ of 0.3 achieved reasonable closure in Polluted-H but resulted in significant over-estimation in Polluted-L by up to 50 to 60% and using bulk $PM_1$ chemical composition enabled to achieve closure in Polluted-H, but showed over-estimation in Polluted-L.

    In Mixed-H sector (Fig. S1), the slopes of regression lines were higher (ranged from 1.1 to 1.6), and the Pearson's R values were lower (0.64 to 0.79). The Mixed sector represents the aerosols of marine modified air masses or continental outflow. As 490     shown in Table S2, the concurrent high concentration of BC and sea-salt and the coexistence of near-hydrophobic and sea-salt mode in GF-PDF confirm their externally mixing in a mixed air mass. Generally, Mixed-H were rather similar to Clean-H, and Mixed-L were more similar to Clean-L, which is expected due to the seasonal distribution of wind direction and synoptic scale circulation at MHD. For the Mixed-L, the $N_{CCN}$ obtained by method D was generally higher than method C but lower than method B, suggesting that dividing GF-PDF into there modes inhibits some information of external mixing, especially the 495     existence of completely non-hygroscopic Aitken particles in the wintertime.

### 4.2.3 The impact of size-dependent hygroscopicity and mixing state

The importance of size-dependent hygroscopicity and mixing state in $N_{CCN}$ estimation has been emphasized in earlier studies. For example, Meng et al. (2014) found the $N_{CCN}$ deviation increasing up to 26% when $N_{CCN}$ was calculated by bulk $PM_1$ chemical composition (without size-dependent hygroscopicity). A CCN closure conducted in winter season in India (Bhattu and 500     Tripathi, 2015) found the closure was sensitive to both aerosol chemical composition and mixing state at lower SS (e.g. 0.18%). While at the higher SS, size-resolved chemical composition and the solubility of organics was needed for accurate $N_{CCN}$ estimation. In this study, the relative difference between each method was established to quantify the effect of simplified size-dependent hygroscopicity and mixing state. The relative difference between method C and method D can be viewed as the impact of size-dependent hygroscopicity. Similarly, the error of ignoring mixing state was obtained by looking at the relative 505     difference between method D and method F.

    As shown in the Fig. 6 (left), the size-dependent hygroscopicity induced relative error increased with at a given SS, and reached the highest value at 0.5% or 0.75% SS, and then decreased again at SS of 1%. The trend was shared by all types of air masses, suggesting the impact of size-dependent hygroscopicity was the most prominent at moderate to high SS values. The assumption of size-independent hygroscopicity may cause larger errors in Clean-L, and the relative differences were within ±25% 510     for Clean-H, and were up to 37% for Clean-L at 0.75% SS. As shown in Table S3, the $κ_{HTDMA}$ of Clean-L increased with particle size, and the trend was similar to previous ship-based observations in the North Atlantic where Aitken mode particles were more hygroscopic than accumulation mode particle because of the higher mass loading of nss-$SO_4$ (Saliba et al., 2020). The relative differences in Clean-L sector increased with increasing SS, which confirmed the fact that Aitken mode particles were less-hygroscopic as would be the case with nss-$SO_4$ vs sea-salt particles. For Mixed and Polluted sectors, the relative differences 515     introduced by size-dependent hygroscopicity were mostly within 25%, ranging from 8% to 25% depending on SS.

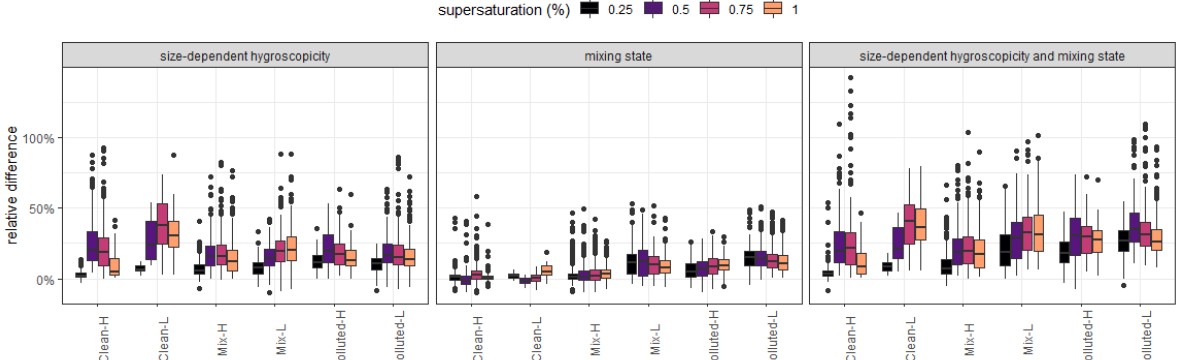

**Figure 6: The relative difference induced by ignoring size-dependent hygroscopicity (left), simplification of the aerosol as complete internal mixing state (middle), total error of the first two assumption (right). The lines represent medians, the boxes represent 25th to 75th quantiles, and whiskers represent 1.5 interquartile and the points represent outliers.**

The impact of simplification of the aerosol population to complete internal mixture on $N_{CCN}$ estimation was well recognized, however not well quantified, Lance et al. (2013) found that neglecting the mixing state lead to overestimation of CCN concentration, during morning rush hour in greater Mexico City. To access the effect of internal mixing, studies often assumed the chemical species being externally mixed (Ren et al., 2018; Zhang et al., 2020). The relative difference between method C and method F provided quantitative assessment of the error introduced by the assumption of internal mixing. As shown in the Fig. 6 (middle), the assumption of complete internal mixing did not induce significant relative difference in clean sector (including Clean-H and Clean-L), due to 1) clean sector aerosols being relatively internally mixed (compared to polluted sector) and 2) the high hygroscopicity of aerosol in clean sector (compared to pure organics). The median relative difference caused by the assumption of complete internal mixing was relatively smaller for Clean-H, suggesting that the Clean-L had a higher degree of external mixing. Such external mixing resulted from the near-hydrophobic particle type in Clean-L as noted several times (Xu et al., 2020). The impact of mixing state varied at different SS. In the Polluted-H, the relative difference induced by the mixing state increased with the increasing SS, while in the Polluted-L the relative difference by mixing state decreased with increasing SS. The seasonal pattern of the impact of mixing state to polluted air masses was similar to a previous study (Wang et al., 2010). As shown in the Fig. 6 (middle), the median relative difference induced by the mixing state was within ±5% for Clean-H, Clean-L and Mixed-H. As expected, the relative difference induced by mixing state was high in Polluted-H and Polluted-L, and Mixed-L. The highest median relative difference of 13% was observed in Polluted-L. The impact of mixing state in polluted sector decreased with increasing SS, which is similar to a previous study conducted in HKUST Supersite (Meng et al., 2014).

A general conclusion can be drawn that the mixing state matters more at low SS while size-dependent hygroscopicity was crucial at higher SS. Since the negligence of size-dependent hygroscopicity and the assumption of internal mixture are commonly adapted practice, the total relative difference was considered as an upper limit for the $N_{CCN}$ error, at least at the current sampling location. As shown in the Fig. 6 (right), the error introduced by the two assumptions was relatively small in clean sector as it resulted in less than 5% deviation between measured and estimated $N_{CCN}$ at low SS (0.25%) in Clean-H, and up to 10% deviation in Clean-L . At higher SS, these two assumptions added up to over 45% error in Clean-L and 20% in Clean-H. On the contrary, it resulted in up to 30% error at low SS and 17% at high SS in the Polluted sector. The effect was less evident with increasing SS, pointing size mattered most at CCN activation.



### 4.2.4 The effectiveness of method E


In principle, method E was a simplified version of method F, which aimed at reducing the computational complexity while preserving the information of mixing state. Besides, the data of these hygroscopic modes were commonly reported in previous studies To evaluate the representativeness of the mixing state by three hygroscopic growth modes versus the mixing state represented by GF-PDF, the relative differences between method E and method F were considered by the following ratio:


$$\frac{N_{CCN\ D} - N_{CCN\ E}}{N_{CCN\ D} - N_{CCN\ F}} \qquad (18)$$

where the $N_{CCN\ D}$ - $N_{CCN\ F}$ is the difference of $N_{CCN}$ between method D and F, which represents the error of $N_{CCN}$ caused by assuming the complete internal mixture while the $N_{CCN\ D}$ - $N_{CCN\ E}$ is the difference of $N_{CCN}$ between method D and E, which represents the $N_{CCN}$ that was corrected by using three hygroscopicity modes. The ratio in the Equation (18) of 100% suggests that three modes can represent the full GF-PDF pretty accurately and a value of 0% suggests the three mode method cannot represents the GF-PDF at all.


As shown in Fig. 7, for the Clean-L, the median values were highest at 1% SS, which is consistent with our previous observations showing that the Aitken mode of marine aerosols in wintertime were external mixture of species. For the Polluted-H and Polluted-L, reducing the full GF-PDF into three modes can diminish the error by up to 50%, when mixing state was completely ignored.


The effectiveness of the method E decreased with the increasing SS, which suggested the increasing fraction of completely hydrophobic species in smaller size ranges. When compares to a CCN closure study in a polluted region in China (Ren et al., 2018), the estimated $N_{CCN}$ using bulk composition is higher than $N_{CCN}$ using size-resolved composition by less than 5% while the assumption of internal mixture induced an error of about 40% to 50%. During the study in Mexico City, Wang et al. (2010) found ~20% deviation by assuming internal mixture and using bulk chemical composition. A study conducted by Wex et al.


(2010) compiled data of different aerosol types and concluded out that for aerosol populations with contribution of less-hygroscopic mode over 50%, the $N_{CCN}$ can be overestimated by 100% by assuming internal mixture. The above study also concluded that the fraction of less-hygroscopic mode is important when the $N_{CCN}$ was calculated by measured aerosol number size distribution and particle hygroscopicity. In the Polluted-L of the current study the number fraction of near-hydrophobic mode particle (GF < 1.3) was lower, leading to a smaller overestimation. The smaller overestimation by assuming complete


internal mixture might due to the lack of hydrocarbon producing sources from the sampling location.



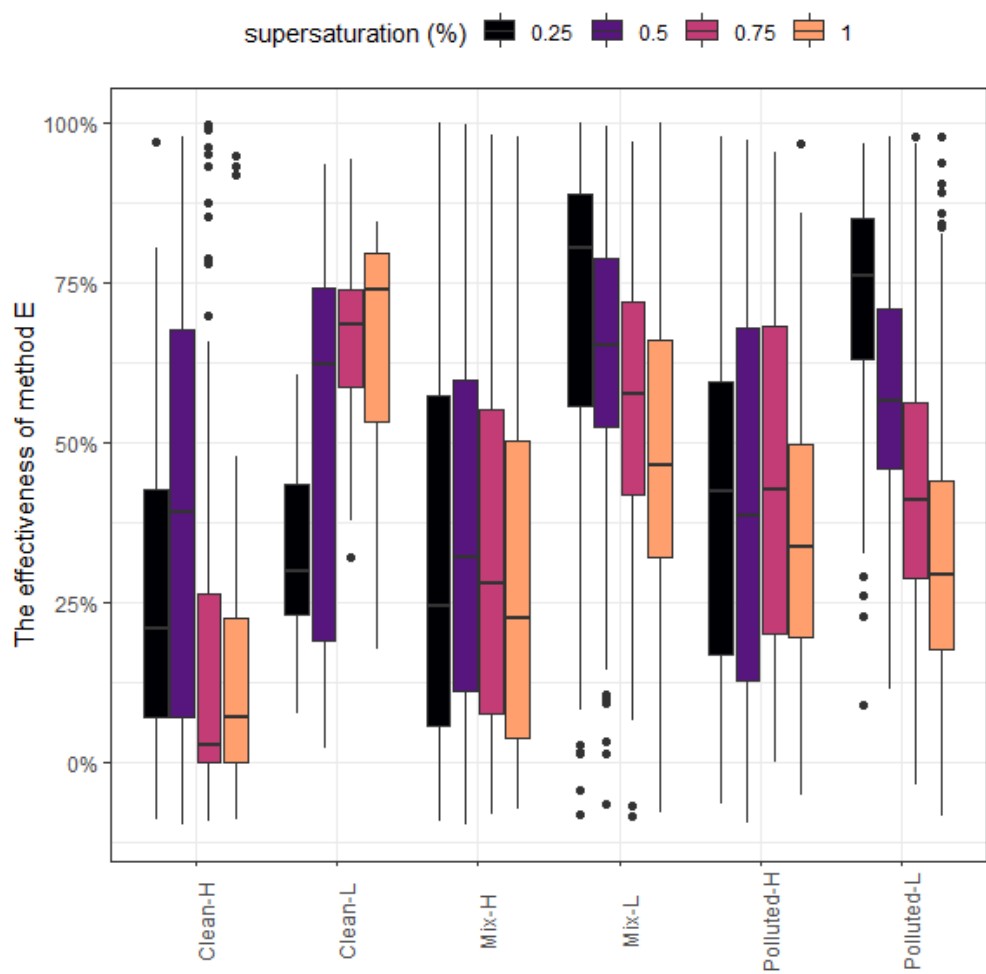

**Figure 7: The representativeness of the three hygroscopic mode method (E) to the real mixing state scenorio (Method F). The lines represent medians, the boxes represent 25th to 75th quantiles, and whiskers represent 1.5 interquartile and the points represent outliers.**

### 5. Conclusion

A reductionist approach is commonly applied to reduce the complexity of aerosol processes and properties. Simplifications, approximations and parametrisations are at the heart of every atmospheric model, however, those should not go beyond realistic representation of aerosol population and should have limited associated uncertainties. Several reductionist methods were applied to calculate $N_{CCN}$ based on different assumptions of mixing state or size dependent hygroscopicity. Generally, good agreement was found using either AMS or HTDMA data and the Pearson's R value between estimated $N_{CCN}$ and measured $N_{CCN}$ ranged

from 0.65 to 0.97 depending on air mass type and the supersaturation level. This study not only provided a quantified evaluation of $N_{CCN}$ estimation based on AMS and HTDMA measurement, but also evaluated the impact of size-dependent hygroscopicity and the assumption of internal mixing on predicting $N_{CCN}$, which led to several key conclusions. Firstly, using a constant κ of 0.3 leads to under-estimation, except in Polluted-L conditions. Bulk $PM_1$ chemical composition and sub-saturated hygroscopicity can be used to estimate $N_{CCN}$ within the range of measurement uncertainty. Secondly, the closure results were poorest in low SS,



therefore careful considerations should be taken at low SS. Thirdly, the impact of size dependent hygroscopicity and mixing state were in terms of the relative error could be as high as 50% in Clean-L and about 25% in polluted sector. The relative difference induced by the assumption of internal mixing was typically less than 25%, due to the lack of nearby fresh emission sources at MHD. Lastly, a reduction of a full GF-PDF representation to the three basic hygroscopicity modes, reduces the error caused by the complete ignorance of mixing state by up to 80%, especially in polluted sector.


*Code and data availability.* The data used in the analysis are published in digital Mendeley repository access through http://doi.org/10.17632/3dx6pnx869.1.

*Author contributions.* WX and KNF conceived the study and analysed the data. JO provided the AMS data. WX, KNF and DC
wrote the manuscript with the input from all the authors.

*Competing interests.* The authors declare no competing interests.

*Acknowledgements.* This work is supported by EPA-Ireland (AEROSOURCE, 2016-CCRP-MS-31), the COST Action CA16109
(COLOSSAL) and supported by MaREI, the SFI Research Centre for Energy, Climate, and Marine [Grant No: 12/RC/2302_P2]. Jakub Bialek is acknowledged for running HTDMA and CCN in Mace Head. The Chinese Scholarship Council (No. 201706310154) is acknowledged for supporting Wei Xu financially.

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
