# Peer review of "The impact of aerosol size-dependent hygroscopicity and mixing state on the cloud condensation nuclei potential over the Northeast Atlantic"

_Atmospheric Chemistry and Physics, 2021_

## Referee Comment (RC2)

This manuscript presents the CCN closure study over the Northeast Atlantic Ocean. By assuming different mixing states and size-resolved $\kappa$ values, the measured and estimated $N_{CCN}$ were compared. I do think the results are quite interesting and important for the model work, as the author points out in the conclusion. However, I strongly recommend the author be more careful about the data evaluation, analysis, and interpretation. The following major comments must be satisfactorily addressed before consideration for publication.

Major comments:

1. In Section 2.2.1, SMPS measured particle number size distribution only covers the particle size up to 500 nm. Once you used the integrated particle number concentration from SMPS-measured size distribution, particle number concentration in the size range larger than 500 nm ($N_{>500nm}$) are not considered. $N_{>500nm}$ are CCN at supersaturations >0.20%. From Fig. 2, even the particle larger than 500 nm is not measured, it is clear that $N_{>500nm}$ cannot be neglected. This is also related to the interpretation of slopes in Fig. 4 & 5.

   One solution is to fit the larger Accumulation mode. Based on such a method, the $N_{>500nm}$ can be estimated.

2. In Section 2.3, the observation was categorized into Clean-H, Clean-L, Polluted-H, Polluted-L and mixture in between. Why did you use the wind direction, rather than be backward trajectory? It is mentioned the pollution is long-range transport anthropogenic pollution from Europe. Therefore, backward trajectories should work better than the wind direction.

   Besides, after classification, it is better to show the wind rose plot; boxplot or frequency distribution of BC, $N_{30}$ and/or meteorology data during Clean-H, Clean-L, Polluted-H, Polluted-L, Mix-H and Mix-L in the supplement. Reviewers and readers will have a better understanding of the classifications. It also helps your interpretation afterward.

   The "H" and "L" are classified by the biological activity seasons; therefore, it is needed to show the difference of biological activity during "H" and "L". For example, you can

show the Chlorophyll-a and DMS concentration over different seasons. There are free and easy access data from NASA.

Why WS>3m s$^{-1}$ is one of the criteria of the clean sector? When BC<15 ng m$^{-3}$, WD from 190 to 300 and WS<3 m s$^{-1}$, will it be classified as which sector?

3. Some of the criteria and numbers are very arbitrary, lack evidence to support.

For example, in Lines 194-201: First, as far as I know, the CPC model 3010 TSI detection limit is 10 nm particles. Second, SMPS upper limit is 500 nm, whereas CPC depends on your inlet cut size. To make it clear, as CPC covered a broad size, N10/Ncpc should be smaller than 1. In a normal distribution, $\mu \pm \sigma$ covers 68% and $\mu \pm 2\sigma$ covers 95%. I could not under why 84.15$^{th}$ quantile. Why "extra 10% of uncertainty was allowed due to typical uncertainty of particle counting"? Here you mean the measurement uncertainty of SMPS?

Lines 203-207: Why are 1.1*N30 and 1.2*N30 used as the limit?

Line 216: Why "uncertainty of the AMS based on $\kappa$ to be lower than 20%"? How do you get 20%?

4. Some of the statements and interpretations are too strong without supporting evidence.

For example, in Line 311, Do you think wet removal is one of the "cloud processing" or not? I am guessing what you are trying to say is that higher concentration of accumulation might be due to strong condensation growth and/or free troposphere entrainment during wintertime.

Lines 327-329: Based on your classification criteria, Polluted and Mix both feature higher BC mass, I presume the air masses during these periods are from the land. Why did Mix-H show greater similarity in number size distribution to Clean-H? Winds during Polluted-L are from 35° to 135°, whereas winds during Mix-L are from 135° to 190° and from 300° to 35°. It is not accurate to say the prevailing winds are similar during Polluted-L and Mix-L.

Minor comments:

Line 152: Since you have not introduced the method F, it is better to delete this sentence here.

Line 210: Please check the unit of BC in Table 1.

Line 306: "Fig. 2 shows" change to "Figure 2 shows".

Line 353: Did you use air mass back trajectory? As suggest in the major comments, the back trajectory is important for your classification and data interpretation.

Lines 366-368: In the classification standard, you only mentioned the wind direction at the measurement site. Wind direction cannot tell the air mass during Clean-L from the ocean rather than from continental. As I said in the major comment, the backward trajectory might be a better classification criterion than wind direction.

Line 452: Change to "Fig. 5".

---

## Author Comment (AC1)

**Reviewer 2**

This manuscript presents the CCN closure study over the Northeast Atlantic Ocean. By assuming different mixing states and size-resolved  $\kappa$  values, the measured and estimated NCCN were compared. I do think the results are quite interesting and important for the model work, as the author points out in the conclusion. However, I strongly recommend the author be more careful about the data evaluation, analysis, and interpretation. The following major comments must be satisfactorily addressed before consideration for publication.

Response: We thank the reviewer for the constructive comments.

Major comments:

1. In Section 2.2.1, SMPS measured particle number size distribution only covers the particle size up to 500 nm. Once you used the integrated particle number concentration from SMPS-measured size distribution, particle number concentration in the size range larger than 500 nm (N>500nm) are not considered. N>500nm are CCN at supersaturations >0.20%. From Fig. 2, even the particle larger than 500 nm is not measured, it is clear that N>500nm cannot be neglected. This is also related to the interpretation of slopes in Fig. 4 & 5. One solution is to fit the larger Accumulation mode. Based on such a method, the N>500nm can be estimated.

**Response:** We acknowledge that the particles larger than 500 nm certainly contributed to CCN. However, their number concentrations were typically very low in the remote atmosphere and normally within the measurement uncertainty of the total number. As shown in the Figure R1 below, the largest particle number concentration with sizes over 500 nm was observed in clean sector with the median N500 of 3.7 particles per cm3. We can then compare with the number of particles above the characteristic Hoppel minimum size - 90 nm in the clean sector - which is arising from the activated particles in the clouds over Mace Head and obtain 145 particles per cm3. Therefore, the fraction of N500 was

Figure R1. The aerosol number size distribution combining scanning mobility particle sizer (SMPS, 0.003 to 0.5 um) and aerosol dynamics sizer (APS, 0.5 to 10 um) for each sector. The lines represent the median values, the shaded area represent  $25^{\text{th}}$  to  $75^{\text{th}}$  quantile. The vertical lines represent SMPS sampling upper size limit of 0.5 um. Also shown are the number of particles above the Hoppel minimum (NH) and over 500nm (N500).

We have added a sentence for clarification in the revised manuscript:

"To note that the upper limit of 500 nm was used when integrating  $N_{CCN}$ , because the SMPS measured particles up to 500 nm. Particles larger than 500 nm certainly contributed to the  $N_{CCN}$  because of their large size. While ignoring those large particles would cause a slight systematic underestimation of  $N_{CCN}$ , such underestimation would be negligible as their contribution to the total activated particle number (<3%) was within the measurement uncertainty of the CCN counter."

2. In Section 2.3, the observation was categorized into Clean-H, Clean-L, Polluted-H, Polluted-L and mixture in between. Why did you use the wind direction, rather than be backward trajectory? It is mentioned the pollution is long-range transport anthropogenic pollution from Europe. Therefore, backward trajectories should work better than the wind direction. Besides, after classification, it is better to show the wind rose plot; boxplot or frequency distribution of BC, N30 and/or meteorology data during Clean-H, Clean-L, Polluted-H, Polluted-L, Mix-H and Mix-L in the supplement. Reviewers and readers will have a better understanding of the classifications. It also helps your interpretation afterward. The "H" and "L" are classified by the biological activity seasons; therefore, it is needed to show the difference of biological activity during "H" and "L". For example, you can show the Chlorophyll-a and DMS concentration over different seasons. There are free and easy access data from NASA. Why WS>3m s-1 is one of the criteria of the clean sector? When BC

---

## Author Comment (AC2)

Reviewer 1:

*This study presents data collected from the Mace Head field operational site and presents an analysis of the influence of the size-dependent hygroscopicity and mixing state of aerosols for the prediction of the CCN number. The authors compare CCN number closure using 6 different methods on aerosol concentrations measured over Northeast Atlantic. The work provides significant long-range measurement of CCN from one site. The sampled air masses are divided into sectors, including polluted continent, clean oceanic, and mixtures for both high and low biological level of activity. It was concluded that for low SS the mixing state plays an important role while for high SS the hygroscopicity is size dependent.*

*The paper is appropriate for submission to ACP. The issue of $N_{CCN}$ closure is of scientific importance and thus the work contributes new data and methods of scientific significance to the field. There are several grammatical errors and typo through the manuscript. There are some questions regarding the approach and interpretation of the analytical methods D-F. There are also additional questions regarding the relevance/reliance on data provided in the supplemental document for the understanding of the main manuscript.*

*Overall, the manuscript and contents continue the important scientific discussion about $N_{CCN}$ prediction and should be published. This reviewer congratulates the authors on synthesizing a significantly large and complex CCN measurement and prediction data set. There are several unique features that can be discerned over a long period of time and the authors have overall presented a cohesive narrative that expands our knowledge of CCN prediction and the aerosol-indirect effect.*

**Response:** We thank the reviewer for the positive comments. We have corrected the grammatical errors and typos, and we clarified our methods and approaches.

*Below are some major and minor comments that are suggested to improve the clarity of the work.*

***Major Comments:***

*A major conclusion (L531) is that the mixing state varies at different SS. However this is mainly true for the Clean and Polluted air masses, as is focused on in the paper. As suggested, these air masses have uniform properties (Table 1). Is the same conclusion true of the mixed air masses? Or is it possible that mixing state impacts the $N_{CCN}$ at both high and low SS in the mixed air datasets? If so, the main conclusion should emphasize to the reader that mixing state has minimal effects on $N_{CCN}$ when the air masses can be separated.*

**Response:** As pointed out by the reviewer, the impact of mixing state varies at different SS, and it is also true for the mixed air masses. Therefore, it is unreasonable to state that mixing state has minimal effect once air masses were separated. As indicated in this study and few other studies in the references (Wex et al., 2010; Schmale et al., 2017), the mixing state has the largest impact when the air masses contained a large number of hydrophobic particles which are often independent of the air mass origin.

*There are concerns with the logic of method C for analysis. Given the authors state that the Aitken mode is that which is most sensitive to changes in hygroscopicity, why choose a size (165 nm) in the accumulation mode? This proliferates to the efficacy of methods for inferring mixing states (Fig 6, middle panel). If method C, is based on a smaller particle diameter (one in the Aitken mode) will it not become more efficient?*

**Response:** The size of 165 nm was selected because the accumulation mode size approximated the bulk PM1 chemical composition (Xu et al., 2020) and, hence, was a logical progression from the Method B where kappa was derived from PM1 bulk chemical composition and applied across all HTDMA sizes. Methods B and C were similar as they derived kappa either from measured chemical composition or the specific size which approximated bulk PM1 mass. The method D was used to isolate the impact of mixing state, while containing the information of size-dependent hygroscopicity.

*Page 16 line 392: The authors claim that there was no statistically significant difference between predictions obtained using method D and method E. However, an analysis for that is not shown. Ideally hypothesis testing by performing p-value estimation or student t-test, the statistical significance of the results can be determined. Statistical significance can probably not be determined simply by comparing the RRSE or R-scores against each other obtained using the 2 methods.*

**Response:** We thank the reviewer for the suggestion. We performed chi-test and added some additional information to the revised manuscript:

"No statistically significant difference was found between method D and method E with the corresponding chi-squared value of 70 (while the critical chi-squared value was 303 at the significance level of 0.05), suggesting that the simplification of full GF-PDF into near-hydrophobic, more-hygroscopic and sea-salt modes was a good representation of aerosol mixing state."

*Page 22 line 560: It does not seem that the effectiveness of method E with respect to methods D and F follow a definitive pattern. The relative error for the continental cases seem to have an almost 0 correlation with the ambient type (polluted, mixed, or clean). And the relative error for the marine cases have both positive and negative correlations with the ambient types. Therefore, on what basis can it be inferred that the effectiveness generally decreases with increasing SS? The explanation seems to be missing or unclear.*

**Response:** It is true that there was no definitive pattern. It is the median effectiveness score that decreased with increasing SS, which prompted our assertion. We have revised the sentence for clarification:

"In general, the median value of the effectiveness of the method E decreased with the increasing SS, which suggested the increasing fraction of completely hydrophobic species in smaller size ranges, except for Clean-L, in which the effectiveness increased with the increasing SS."

*The use of both the Pearson score and RRSE is unclear. How are the RRSE values interpreted and what do they represent for individual closures? Pearson's coefficient is self-sufficient for quantifying the uncertainties between measured and estimated/predicted values of a variable, then what purpose are RRSE scores serving?*

**Response:** Pearson coefficient may not be able to capture the discrepancy between the measured and predicted

values. For example, a high Pearson R value suggests good correlation (as long as it is statistically significant), but not necessarily suggests good closure, as there can be some systematic under- /over- estimation. The RRSE scores can quantify the absolute difference (error) between the measured and estimated Nccn. The RRSE was particularly useful when the slope was close to 1, but R was not close to 1. Statistical significance of the correlation with specific P-value would serve similar purpose, but we chose a more intuitive RRSE.

*This reviewer has some reservations about the calibration of the instruments. How often was the CCNC calibrated? The frequency of calibration is also associated with the +/- 0.03% potential SS drift; the longer the times between calibration the greater potential for greater SS uncertainty. Furthermore, it may be of interest to include an explanation of how the HTDMA setup and RH were calibrated/validated as the derived hygroscopicity is applied to 3 of 6 methods. The conclusion can also benefit with a summary of the advantage/disadvantages of each method.*

**Response:** The reviewer's concerns are well founded. Although calibration of CCN and HTDMA were regularly performed, deviations from calibrations were inevitable. Frequent calibrations do not necessarily ensure fully trustworthy dataset. Therefore, we invested a lot of effort into data validation procedures, which limited our dataset, but increased data credibility.

We added additional information on the instrumentations:

"The HTDMA RH probes was calibrated using ammonium sulfate monthly, the sizing of HTDMA were regularly cross-checked with SMPS, and the sizing difference between two DMAs were performed regularly using dry-scan."

The summary of each method is now added in the revised manuscript.

"Method A was the simplest method that can provide a rough estimate of $N_{CCN}$ if hygroscopicity data was not available. Method B was probably the most frequently used in the scientific community because of various parallel on-line and/or off-line aerosol chemical analyses. Method C was the simplest version of hygroscopicity closure and the complexity increased with the Methods D, E and F. The Method F was the most comprehensive method that capture the fine structure of GF-PDF, but which is not always presented in the previous studies and is computationally expensive. Method E, therefore, divided full GF-PDF into three discrete modes and reduced the complexity of GF-PDF, but kept basic information of the mixing state. More importantly, the principal hygroscopicity modes used in Method E were widely reported in previous HTDMA studies, which then can be readily used for the Nccn prediction."

*Page 7 line 215: The criteria for considering the uncertainty of 20% in the hygroscopicity seems unclear. Specifically, how do the stated uncertainties in the other measurements relate to the assumed uncertainty of 20% in ? Is there a mathematical explanation behind this, or was this empirically chosen based on the findings of the authors or previous studies?*

**Response:** The main uncertainty of AMS was determined by collection efficiency, which is approximately 30-40% (Bahreini, R. et al). However, for determining the relative contribution of chemical species, which matters most for kappa calculation, the uncertainty is smaller. Moreover, it was demonstrated that the kappa values from AMS were usually within 20% from HTDMA derived GF (Ovadnevaite et al., 2017; Xu et al.,

2020). Therefore, we considered the uncertainty of chemical composition derived kappa based on AMS was within 20%.

*The author, throughout the paper, makes many repeated references to figures and tables that are found in the supplemental section. Considering their importance, wouldn't it be more appropriate to incorporate them in the paper itself?*

**Response:** Thank you for a logical suggestion. We have moved the most frequently referred Table S2 and S3 to the main text. However, the rest of the supporting figures mostly referred to specific methods and are superfluous to the main text.

*Lastly, a summary lookup table of the effects of each method would be useful to have in the main paper. This can be appended onto Table 3 in a column that shows the relative error, or approximate magnitude of over/underestimation.*

**Response:** We have appended the median relative error of each Method in Table 3 (without distinguishing air masses and supersaturation levels).

And we have added an additional sentence for further clarification:

"The median relative error without distinguishing air masses and supersaturation is showed in Table 3"

**Table 3: The simplification and assumption of $N_{CCN}$ estimation methods.**

| Method | Hygroscopicity proxy | Mixing state assumption | Size dependent hygroscopicity | Temporal variation | Median relative error |
|--------|----------------------|-------------------------|-------------------------------|--------------------|-----------------------|
| A | constant $\kappa$ of 0.3 | internal mixing | no | no | 0.015 |
| B | bulk PM1 chemical composition | internal mixing | no | yes | 0.17 |
| C | mean GF | internal mixing | no | yes | 0.11 |
| D | mean GF | internal mixing | yes | yes | -0.04 |
| E | number fractions of | quasi | yes | yes | -0.06 |

| | | hygroscopicity modes | external mixing | | | |
|---|---|---|---|---|---|---|
| F | GF-PDF | external mixing | yes | yes | -0.09 |

**Minor Comments:**

*Line 15-16: Please check method description. "Method C utilized size dependent…" Isn't this D?*

Corrected.

*Line 27: GF-PDf or GF-PDF?*

Corrected.

*Line 33: supersaturation vs super-saturation (line 11)*

Corrected.

*Line 51: nss-sulfate is not defined until line '154.*

Corrected.

*Line 147:   arbitrary? Mixing State can be quantified. Please see Riemer, N., et al*

**Response:** We thank the reviewer for pointing this out. Actually, we cited the study using the same metric for mixing state. The mixing state was indeed parameterized, however, it can only be obtained based on single particle chemical composition, which is not available in our study.

We have added a sentence for clarification:

"It is important to note that the definition of mixing state is arbitrary, although it was recently defined as the heterogeneous distribution of chemical species across the aerosol populations (Riemer et al., 2019; Ching et al., 2017). However, the mixing state metric proposed by Riemer et al. (2019) can only be obtained from single particle chemical composition, which is not available in this study."

*Line 159: the letter d is missing from "and", "ammonium nitrate an sulfate"*

Corrected.

*Line 188: Please be aware that the font size and notation in the subscripts is not consistent. Similar inconsistencies can be found on other pages as well (pg 7, 8, so on and so forth).*

Corrected throughout the paper.

*Table 1: It is not clear in the criteria in the table if it is and/or. For example, BC< 15 ng AND WD from? Or is it one or the other?*

**Response:** The table is representing "and".

We have modified the table for clarification.

*Line 205: Please reword "were taking into". Take into?*

**Response:**

We have modified the sentence:

"The different upper limits were used due to the larger uncertainty in SMPS measurement at low total particle number concentration."

*Page 8: The alignment of the columns in Table 2 is awkward. Perhaps centering?*

Corrected.

*Table 2: Is a reference for the density of organics required? What is the justification for 1400.*

The density of organics was chosen to represent oxidized organics in aged air masses (Alfarra et al., 2006), which dominate at remote locations. The reference was added.

*Line 239-240 why is the upper limit 500nm? Is 500nm the limitation in the DMA measurement?*

**Response:** The 500 nm size was the upper limit of our SMPS system comprising of standard narrow DMA column (Vienna-type as opposed to Hauke-type used in HTDMA).

*Line 269: Change "In method F" to "In method E".*

Corrected.

*Line 339 lend support "to"?*

Corrected.

*Line 343: Typo. Change to "Respectively".*

Corrected.

*Line 343: HB and LB not defined.*

Corrected to H and L.

*Line 355: "Fig 4&-5"?*

Corrected.

*Line 356: This range is peculiar. The minimum value of Pearson' coefficient does not seem to be 0.85 (more like 0.76 in fig. 4 and 0.65 in fig. 5).*

*The range was calculated for the whole dataset rather than specific sector,*

We have added additional information for clarification:

Overall, the estimated and measured $N_{CCN}$ agreed well and were highly correlated with the Pearson's R ranging between 0.85 and 0.99 for the whole dataset.

*Line 361: Sentence is unclear.*

We rewrote the sentence as suggested:

"The estimated $N_{CCN}$ using different methods are summarised in Table S1."

*Line 373, 380-381: Please double check ranges. The slope and R minimum and maximum values mentioned here do not match the ones written on the panels in fig. 4.*

Corrected.

*Line 423: The word "eith" is likely misspelt.*

Corrected.

*Line 454 "That indicated the potential over-estimation of 30% to 50% by using κ of 0.3 in highly polluted air masses" Is kappa or Nccn overestimated 30~50%?*

**Response:** It was the Nccn that was overestimated when using a constant kappa of 0.3.

We modified the sentence for clarification.

*"That suggested that using κ of 0.3 may induce the potential over-estimation of 30% to 50% in highly polluted air masses."*

*Line 473: Change "compare" to compared*

Corrected.

*Line 485 "To conclude, using κ of 0.3 achieved reasonable closure in Polluted-H but resulted in significant over-estimation in Polluted-L by up to 50 to 60% and using bulk PM1 chemical composition **enabled to achieve** closure in Polluted-H, but showed over-estimation in Polluted-L."  What is enabled to achieve? Simply achieved?*

**Response:** The sentence has been modified.

"To conclude, by using constant κ of 0.3 reasonable closure in Polluted-H was achieved, but resulted in significant over-estimation in Polluted-L by up to 50 to 60%; and by using bulk PM1 chemical composition derived κ again resulted in reasonable $N_{CCN}$ prediction in Polluted-H, but showed over-estimation in Polluted-L."

*Line 539: verb tense? Commonly adapted practice(s).*

Corrected.

*line 541: Please rephrase 'pointing size mattered most at CCN activation.' Indicating that?*

**Response:** The sentence was, indeed, unclear and was modified:

"The error decreased with the increasing SS,  pointing to the fact that the size plays a more important rule role in CCN activation at high SS."

*Line 589: Reduced the error by 80% with respect to what?*

**Response:** The statement was with respect to assuming complete internal mixing.

The sentence has now been revised for clarification.

Lastly, a reduction of a full GF-PDF representation to the three basic hygroscopicity modes, reduced the error caused by assuming complete internal mixture (Method D) by up to 80%, especially in polluted sector.

Reference:

Alfarra, M. R., Paulsen, D., Gysel, M., Garforth, A. A., Dommen,J., Pŕevôt, A. S. H., Worsnop, D. R., Baltensperger, U., and Coe,H.:  A mass spectrometric study of secondary organic aerosols formed from the photooxidation of anthropogenic and biogenic precursors in a reaction chamber, Atmos. Chem. Phys., 6, 5279–5293, 2006.

Bahreini, R., Ervens, B., Middlebrook, A. M., Warneke, C., Gouw, J. A. de, DeCarlo, P. F., Jimenez, J. L., Brock, C. A., Neuman, J. A., Ryerson, T. B., Stark, H., Atlas, E., Brioude, J., Fried, A., Holloway, J. S., Peischl, J., Richter, D., Walega, J., Weibring, P., Wollny, A. G., and Fehsenfeld, F. C.: Organic aerosol formation in urban and industrial plumes near Houston and Dallas, Texas, J Geophys Res Atmospheres 1984 2012, 114, https://doi.org/10.1029/2008jd011493, 2009.

Ovadnevaite, J., Zuend, A., Laaksonen, A., Sanchez, K. J., Roberts, G., Ceburnis, D., Decesari, S., Rinaldi, M., Hodas, N., Facchini, M. C., Seinfeld, J. H., and C, O. D.: Surface tension prevails over solute effect in organic-influenced cloud droplet activation, Nature, 546, 637 641, https://doi.org/10.1038/nature22806, 2017.

Riemer, N., Ault, A. P., West, M., Craig, R. L., and Curtis, J. H.: Aerosol Mixing State: Measurements, Modeling, and Impacts, Rev Geophys, 57, 187 249, https://doi.org/10.1029/2018rg000615, 2019.

Schmale, J., Henning, S., Decesari, S., Henzing, B., Keskinen, H., Sellegri, K., Ovadnevaite, J., Pöhlker, M. L., Brito, J., Bougiatioti, A., Kristensson, A., Kalivitis, N., Stavroulas, I., Carbone, S., Jefferson, A., Park, M., Schlag, P., Iwamoto, Y., Aalto, P., Äijälä, M., Bukowiecki, N., Ehn, M., Frank, G., Fröhlich, R., Frumau, A., Herrmann, E., Herrmann, H., Holzinger, R., Kos, G., Kulmala, M., Mihalopoulos, N., Nenes, A., O'Dowd, C., Petäjä, T., Picard, D., Pöhlker, C., Pöschl, U., Poulain, L., Prévôt, A. S. H., Swietlicki, E., Andreae, M. O., Artaxo, P., Wiedensohler, A., Ogren, J., Matsuki, A., Yum, S. S., Stratmann, F., Baltensperger, U. and Gysel, M.: Long-term cloud condensation nuclei number concentration, particle number size distribution and chemical composition measurements at regionally representative observatories, Atmospheric Chemistry and Physics, 18(4), 2853–2881, doi:10.5194/acp-18-2853-2018, 2018.

Wex, H., McFiggans, G., Henning, S. and Stratmann, F.: Influence of the external mixing state of atmospheric aerosol on derived CCN number concentrations, Geophysical Research Letters, 37(10), L10805, doi:10.1029/2010gl043337, 2010.

Xu, W., Ovadnevaite, J., Fossum, K. N., Lin, C., Huang, R.-J., O'Dowd, C., and Ceburnis, D.: Aerosol hygroscopicity and its link to chemical composition in the coastal atmosphere of Mace Head: marine and continental air masses, Atmos Chem Phys, 20, 3777–3791, https://doi.org/10.5194/acp-20-3777-2020, 2020.